# The Bulb, the Brain and the Being: New Insights into Olfactory System Anatomy, Organization and Connectivity

**DOI:** 10.3390/brainsci15040368

**Published:** 2025-03-31

**Authors:** Anton Stenwall, Aino-Linnea Uggla, David Weibust, Markus Fahlström, Mats Ryttlefors, Francesco Latini

**Affiliations:** 1Department of Medical Sciences, Uppsala University, 751 85 Uppsala, Sweden; anton.stenwall@neuro.uu.se (A.S.); aino.ugglakalvas@stockholmssjukhem.se (A.-L.U.); dafvor@hotmail.com (D.W.); mats.ryttlefors@akademiska.se (M.R.); 2Department of Radiology and Nuclear Medicine, Uppsala University, 751 85 Uppsala, Sweden; markus.fahlstrom@uu.se

**Keywords:** olfactory system, white matter dissection, connectivity, tractography, cognition

## Abstract

Background/Objectives: Olfaction is in many ways the least understood sensory modality. Its organization and connectivity are still under debate. The aim of this study was to investigate the anatomy of the olfactory system by using a cadaver fiber dissection technique and in vivo tractography to attain a deeper understanding of the subcortical connectivity and organization. Methods: Ten cerebral hemispheres were used in this study for white matter dissection according to Klingler’s technique. Measurements of different cortical structures and interhemispheric symmetry were compared. Diffusion tensor imaging sequences from twenty-five healthy individuals from the Human Connectome Project dataset were used to explore the connectivity of the olfactory system using DSI Studio. White matter connectivity between the following were reconstructed in vivo: (1) Olfactory bulb to primary olfactory cortices; (2) Olfactory bulb to secondary olfactory cortices; (3) Primary to secondary olfactory cortices. The DTI metrics of the identified major associative, projection and commissural pathways were subsequently correlated with olfactory function and cognition in seventy-five healthy individuals with Spearman’s rank correlation and the Benjamini–Hochberg method for false discoveries (CI 95%, *p* < 0.05) using R. Results: 1. The dissection showed that the lateral stria was significantly longer on the left side and projected towards the amygdala, the entorhinal and piriform cortex. 2. The medial stria was not evident as a consistent white matter structure. 3. Both dissection and tractography showed that major associative white matter pathways such as the uncinate fasciculus, the inferior fronto-occipital fasciculus and cingulum supported the connectivity between olfactory areas together with the anterior commissure. 4. No significant correlation was found between DTI metrics and sensory or cognition test results. Conclusions: We present the first combined fiber dissection analysis and tractography of the olfactory system. We propose a novel definition where the primary olfactory network is defined by the olfactory tract/bulb and primary olfactory cortices through the lateral stria only. The uncinate fasciculus, inferior fronto-occipital fasciculus and cingulum are the associative pathways supporting the connectivity between primary and secondary olfactory areas together with the anterior commissure. We suggest considering these structures as a secondary olfactory network. Further work is needed to attain a deeper understanding of the pathological and physiological implications of the olfactory system.

## 1. Introduction

The olfactory system together with hearing, vision and taste represent the special sensory systems of the human body. It is highly evolutionarily conserved and represents the oldest known sensory system for environmental awareness in living organisms [1,2]. The olfactory system is crucial to survival in many different animal species, enabling and facilitating adequate interaction with the surroundings through itself and through integration with other sensory inputs [2].

Olfaction is involved in and influences many different aspects of human life, ranging from gustatory sensations and situational awareness to hormonal responses and mental health [3,4,5].

Partial or complete loss of olfactory function has multiple consequences for the afflicted individual, and olfactory dysfunction is associated with reduced quality of life affecting a wide range of daily life aspects such as the ability to taste food and beverages and the ability to detect unpleasant or harmful odors [3,5], general wellbeing [6] and cognitive decline [7]. Further research into the possible connection between olfactory input, cognition and personality has shown positive correlation between the ability to detect odors and personal agreeableness derived from the five-factor personality model, a result that suggests that olfactory processing can be influential in complex cognitive procedures such as social interaction and personality traits [8] and behavioral choices [9]. The five-factor personality model is an organizational model for personality traits based on five key terms: extraversion, agreeableness, neuroticism, conscientiousness and openness to experience [10]. Interestingly, olfactory stimulation has been evaluated as a potential therapeutic modality for depressive disorders [11].

The literature clearly demonstrates that intentional odorants can negatively or positively impact cognitive function depending on the stimuli in both individuals with sensory loss or with intact olfactory functions [12]. Therefore, there is a growing evidence that olfactory training/stimulation not only improves olfaction but also positively affects the gray matter volume of several supratentorial or infratentorial areas [13].

The anatomical description of human cortical olfactory areas are traditionally divided into primary and secondary olfactory regions where the primary regions receive input from the olfactory bulb, and the secondary regions receive input relayed from the primary olfactory regions [14]. The axons of the tufted cells and the mitral cells coalesce ipsilaterally to form the olfactory tract on each side, which, in turn, form a medial and a lateral stria [15]. The striae projects to several cortical areas in the medial temporal lobe as well as the basal aspect of the frontal lobe. These areas are classically designated the primary olfactory areas [16]. These areas consist of the anterior olfactory nucleus and the olfactory tubercle, the piriform cortex, the olfactory tubercle, parts of the amygdala and the rostral entorhinal cortex [17]. Thus, the secondary olfactory areas consist partly of limbic and paralimbic brain areas that are involved in regulating behavior, memory and cognition.

Numerous studies have attempted to describe the gross anatomy of the olfactory system in different mammals with different techniques [18,19,20]. In some other cases, the olfactory disfunction has been the target of investigation with new radiological findings and connectivity aspects [21].

Despite all the efforts in describing this system with ex vivo and in vivo techniques, some anatomical aspects are still under debate. For instance, the termination of the stria (medial and lateral) is not clearly studied.

The aim of this study is to investigate the anatomy of the human olfactory system through white matter fiber dissection of human cadaver brains and compare the results with in vivo tractographic data from Human Connectome Project (HCP) subjects. We also utilized sensory and cognitive assessment data from the HCP to elucidate if there are any correlations between diffusion tensor imaging (DTI) metrics and sensory function such as odor detection and taste, and assessed cognitive function.

## 2. Materials and Methods

### 2.1. Human Brain Specimen

Ten human cerebral hemispheres, five right-sided and five left-sided, were provided by The Department of Medical Cell Biology, Uppsala University (Uppsala, Sweden), and included in the study. The specimens came from 8 cadavers (4 females and 4 males, age 49–89 yrs, mean age 76 yrs). Every donor included in the study had given written consent for the use of tissue for educational and research purposes, and the donors were included in the study under ethically approved premises granted by the Regional Ethical Review Board, Uppsala (Dnr 2014/468).

### 2.2. White Matter Dissection

The tissue preparation protocols utilized in this study have been previously described by Latini et al. in detail [22]. The brains were initially fixed through an intra-carotid injection with 12% formalin within one week from the date of death. The brains were subsequently removed and put in 10% formalin for 24 h after which the pia mater, the arachnoid membranes and the vascular structures were carefully removed. The brains were then frozen in −15° for 6–10 days and defrosted for 12 h. Prior to dissection, cortical structures were measured and documented. Each hemisphere was subsequently dissected in a careful stepwise manner under microscopic magnification, working from the lateral to the medial aspect, and from the superficial layers towards the subcortical layers from the baso-frontal and temporobasal regions. The morphological properties of several structures were measured during the dissection procedure (Figure 1A–D), and we calculated the mean, range and standard deviation values as presented in Results: Table 1 and Table 2. We assumed the measurement values to be normally distributed and performed an unpaired two-tailed *t*-test to elucidate any significant differences between the left and right length and width of the olfactory tract and lateral stria as presented in Table 2.

### 2.3. HCP Database Subjects

Post-processed diffusion and T1 data from 100 subjects aged between 26 and 35 were extracted from the WU-Minn HCP database (the 1200 Subjects Data release, https://db.humanconnectome.org) [23]. Post-processing included distortion correction, eddy and motion correction, gradient non-linearity correction and calculation of b-value/b-vector deviation [24,25,26]. The age group was chosen to represent the fully matured human brain.

### 2.4. Multimodal Testing Data

We included multimodal testing data covering olfactory function (NIH Toolbox odor identification test and NIH Toolbox taste intensity test), cognition, emotion and personality traits for each individual. The specific parameters were evaluated using the NIH Toolbox for Assessment of Neurological and Behavioral Function. The emotional and cognitive parameters were evaluated using the emotional and cognitive specific parts of the Toolbox testing protocol, respectively. Personality traits were assessed using the five-factor model of personality model (NEO-FFI). The relevant testing protocols can be reviewed at https://db.humanconnectome.org.

### 2.5. Fiber Tracking Analysis

The fiber tracking analysis was performed in DSI Studio 4 software through a deterministic approach which utilizes a generalized fiber tracking method [27]. For the tracking procedure, we selected a 0.2 mm step size, minimum fiber length 20 mm and a turning angle threshold of 45°. For progression locations containing >1 fiber orientation, the fiber orientation most congruent with the incoming direction and <45° turning angle was selected to indicate subsequent direction. Each progressive voxel’s moving directional estimate was weighted by 20% of the incoming directions of the previous voxels and by 80% of its nearest fiber orientation. This sequence was then repeated to generate fiber tracts. Termination of the tracking algorithm was initiated when the quantitative anisotropy dropped below a subject-specific value, when fiber tract continuity failed to meet the progression criteria or when 100,000 tracts were generated [27]. We selected a QA termination threshold between 0.02 and 0.08 by considering the number of false continuities generated in each individual dataset and choosing a value that enabled optimal detail with minimal noise and a smoothing parameter value of 50% [28]. We set minimum fiber length to 10 mm instead of the common value of 20 mm as some of our areas of interest are in close proximity to each other.

### 2.6. In Vivo Explorative White Matter Analysis

We divided the whole cohort into 2 subgroups: cohort 1 (25 subjects) used for exploratory analysis, used to replicate the white matter dissection results, and a second cohort, cohort 2 (75 subjects) used for quantitative analysis of DTI parameters and correlation group.

### 2.7. Regional-Based Analysis (Cohort 1)

We chose the primary olfactory regions as defined by the literature (the entorhinal cortex, the piriform cortex and the amygdala) as the region of interest. The secondary olfactory regions were defined as the medial orbitofrontal cortex, the cingulate cortex, the insula and the hippocampus. All regions of interest were derived in each individual from the FreeSurfer cortical parcellations map (from the FreeSurfer in-built atlas in DSI studio [29]), except for the Kleist functional area described above and the piriform cortex, which was derived from the Kleist atlas [30]. We also manually created an ROI for the olfactory bulb, but it is not defined in any of the available atlases in DSI Studio. Due to the inherent difficulty in localizing frontobasal structures on MRI sequences (T1 weighted sequence from each subject), we were unable to reliably include the proximal bulb in our volumetric analysis. All cortical areas included in our analysis are represented in Figure 2.

To understand which white matter tracts were closely related to the olfactory regions, we performed the tractographic reconstruction of the olfactory system in a stepwise manner. We started by drawing the olfactory bulb bilaterally into MNI space. Then, we selected the primary olfactory areas (the entorhinal cortex, the piriform cortex and the subcortical amygdala) followed by the secondary areas (the orbitofrontal cortex, the insula, the hippocampus and the cingulate cortex). We proceeded by analyzing the connectivity between the olfactory bulb and each primary olfactory area per hemisphere in each subject followed by the connectivity between the bulb and each secondary olfactory area. For each tract, we analyzed both qualitative (tract designation, tract type) and quantitative (number of tracts, mean tract length, tract diameter, tract volume, fractional anisotropy, mean diffusivity, axial diffusivity and radial diffusivity) parameters. The most anterior and rostral part of the cingular cortex was chosen to represent the cingulum, and we defined the medial and lateral orbitofrontal cortices as one singular unit. We used the automatic recognition of white matter tracts in DSI Studio as a guide for identification of the major pathways and proceeded by performing a visual and anatomical control after the initial automatic recognition.

### 2.8. Atlas-Based Analysis (Cohort 2)

Some major associative, projection and commissural pathways were consistently identified in both white matter dissection and exploratory connectivity analysis. The white matter bundles were then reconstructed on HCP1065 template in MNI space and used for quantitative analysis in GQI-derived space for each subject. The tracts were reconstructed following the same technique used to build the Brain Grid atlas [31]. To decrease the number of possible correlations and confounding factors, the identified major white matter tracts from the explorative stage were selected for a second step analysis for the correlation with sensory and cognitive tests.

### 2.9. Statistical Analysis of Tractography Data and Multimodal Testing Scores

We performed successive correlation analyses between the mean DTI tract metrics (QA, FA, MD, RD and AD) of the AC, the left and right IFOF, UF and UC of cohort number 2, and the results from the odor identification test and taste intensity test from the NIH Toolbox assessment. We subsequently expanded our analysis screening for possible correlations within the olfactory system by adding correlation analyses for cognitive parameters from the NIH Toolbox assessment. We also included the tracts representing olfactory projection fibers from the lateral stria in the correlation analysis. Correlations between tract metrics and the toolbox assessment parameters were calculated using Spearman’s rank correlation coefficient. All results were successively evaluated using the Benjamini–Hochberg method for false discoveries. Correlations with a *p*-value < 0.05 and an FDR value < 0.05 were considered significant. In order to elucidate any trends towards olfactory lateralization, we calculated the symmetry coefficient of the tract metrics QA, FA, MD, RD and AD of the right and left hemispheres according to the formulae (Mean left − Mean right)/(Mean left + Mean right) where negative results indicate lateralization to the right and vice versa, as described earlier [32]. All statistical analysis was performed in R Studio version 2023.09.1+494.

## 3. Results

### 3.1. Ex Vivo Measurement of White Matter Structures

The morphological properties of the olfactory system white matter tracts are presented in Table 1. We found a significant difference in the length of the lateral stria between the left and right hemispheres as described in Table 2.

### 3.2. Ex Vivo White Matter Dissection and Analysis

After the initial measurement, the dissection procedure begun by dissecting the olfactory bulb areas (Figure 3A–D). The bulb, in general, was an elliptic flat structure on the ventral surface of the frontal lobe resting in the olfactory sulcus. The bulb continued caudally as the tract which was more firm and lighter-colored. The tract was isolated from adjacent structures, and in eight out of ten cases, it was associated with gray matter in the distal part of the gyrus rectus and intentionally preserved while the rest of the gyrus rectus was removed. Further distally, the gray substance around the olfactory striae was carefully peeled away, further exposing the striae. The results of the dissection procedure of the olfactory area varied greatly depending on specimen texture and preservation, being distinct in some cases and heavily intermingled with surrounding tissue in others. In three specimens, the lateral striae were in the same superficial plane as the adjacent gray matter, making it easily identifiable but prone to damage. The pathways of the striae varied between the subjects, and it was difficult to differentiate them from surrounding gray matter in seven specimens. In these cases, we removed the cortex above the anterior perforated substance to expose the fibers projecting laterally from the olfactory trigone. In most cases, at least one prominent projection could be identified, usually projecting towards the amygdala. Following the medial olfactory stria distally, we noted that it was frequently less evident than the lateral stria. We could follow the medial stria in eight specimens, from the olfactory trigone towards the corpus callosum where it curved along the subcallosal gyrus. In four specimens, the medial stria could not be reliably identified. In eight specimens, we identified fiber projections rostrally from the olfactory trigone, seemingly originating from the trigone itself. These projections represented a direct connection from the bulb to the frontal lobe which has not been previously demonstrated (Figure 4A–C).

We subsequently proceeded to the inferior gyri of the temporal lobe where the cortex of the parahippocampal gyrus and the fusiform gyrus was removed and proceeded with both a standard lateral dissection to show associative pathways and insular cortex and an inferomedial approach as previously described to display the connectivity of the fusiform gyrus and the parahippocampal gyrus (Figure 5A–D). In order to achieve full exposure, we peeled away the white matter above the hippocampus. The lateral ventricle was followed and opened anteriorly, removing white matter until the caudal amygdala became visible. We were able to locate the inferior longitudinal fasciculus in the fusiform gyrus and the superior longitudinal fasciculus in the operculum and the insular area.

We then dissected the frontobasal surface by removing gray substance in the ventral and medial parts of the frontal lobes beginning in the orbital gyri and continuing medially to the cingulate gyrus. The U-fibers of the orbital gyri were peeled away, and we could uncover the fibers of the uncinate fasciculus and the inferior fronto-occipital fasciculus. Fibers from the anterior cingulate gyrus and anterior commissure were consistently identified as closely related to respectively the medial or lateral stria. A consistent close relationship between IFOF, UF and primary and secondary olfactory areas was identified.

### 3.3. Tractographic White Matter Analysis

We performed a stepwise tractography analysis based on the results from the white matter dissection, exploring three levels of connectivity within the olfactory system in the cohort 1:(1)Tracts from the olfactory bulb/tract to primary olfactory areas (Figure 6A);(2)Tracts from the olfactory bult/tract to secondary olfactory areas (Figure 6B);(3)Tracts from the primary olfactory areas to secondary olfactory areas (Figure 6C).

The first step showed that fibers resembling those from lateral stria could be identified as a projection pathway. Fibers originating from the bulb and directed to the frontobasal cortex were also noted but interpreted as a medial part of the uncinate fasciculus. Thus, the uncinate fasciculus, the anterior commissure and the inferior fronto-occipital fasciculus were the most represented pathways connecting the bulb/tract to the primary olfactory areas. The second step of analysis displayed the uncinate Fasciculus, anterior commissure, inferior fronto-occipital fasciculus and cingulum parolfactory as the most important connections, also including corpus callosum tapetum and, to a lesser extent, Fornix. The most represented pathways between the primary and secondary olfactory areas included uncinate fasciculus, anterior commissure, inferior fronto-occipital fasciculus, cingulum parahippocampal, left Fornix and cingulum frontal parahippocampal. Cingulum, parahippocampal parietal, inferior longitudinal fasciculus, Extreme Capsule and corpus callosum tapetum were also noted, but to a lesser extent (Table 3, Figure 6). Raw data from the explorative analysis om cohort 1 can be found in Appendix A.

### 3.4. Sensory and Cognitive Correlations (Cohort 2)

Our statistical analysis showed no significant correlations (*p* < 0.05, BH FDR > 0.05) between DTI metrics and olfactory function or taste as assessed in the HCP dataset. We found no significant correlations (*p* < 0.05, BH FDR > 0.05) between DTI metrics and cognitive functions previously described to be positively associated with olfactory stimulation. We found positive correlations between the axial diffusivity, the mean diffusivity and the radial diffusivity, and higher scores on the age-adjusted oral reading recognition test from the HCP cognitive test package. A complete list of the statistical findings can be found in the Appendix A.

## 4. Discussion

Despite being one of the special senses of human beings, the olfactory system is not yet sufficiently investigated with regard to its intrinsic connectivity. Much of the available literature on its anatomy is based on relatively sparse dissection data and conclusions drawn from animal studies. A strong body of research points towards its possible involvement in complex cognitive processing. Our work showed three main results:

(1) We found new insights into the anatomy and symmetry of the medial and lateral stria as the dissection showed a significantly longer stria on the left side projecting towards the amygdala, the entorhinal and piriform cortex while the medial stria was not a consistently present white matter structure.

(2) We found that IFOF, UF, AC and anterior cingulum are consistently represented pathways in relation to the olfactory system in both white matter dissection and tractography.

(3) No significant correlations were found between DTI parameters of the lateral stria, IFOF, AC, cingulum, UF or the olfactory tracts, and odor identification/taste intensity test results.

### 4.1. Olfactory System Connectivity: Primary Olfactory Network

One key aim of our work was to investigate the structural connectivity of the olfactory system. We identified a primary olfactory network with the olfactory tract-bulb connecting through the striae to primary olfactory areas on the temporal lobe.

In accordance with the literature, we found that the axons from the olfactory tract project towards the primary olfactory areas: the olfactory tubercle, the anterior olfactory nucelus, subregions of the amygdala, the rostral entorhinal cortex and the piriform cortex [33,34,35]. The lateral stria was partly difficult to discern macroscopically, as previously described [36]. In our material, the lateral stria either had a single bundle or branched into two bundles as evident in four specimens. In two cases, two of these branches projected towards the uncus and the entorhinal cortex, as previously described [37]. The remaining projections continued seemingly towards the amygdala. We found white matter fibers projecting laterally away from the olfactory trigone along the lateral olfactory stria in most specimens. These fibers, despite a very thin appearance, were seen radiating towards the area of the piriform cortex. Thus, we report the lateral stria projecting towards the amygdala, the entorhinal cortex and the piriform cortex, three out of five structures previously described as primary olfactory cortices.

The medial stria was less evident in general than the lateral stria, and we were unable to reliably isolate it in a few specimens. In some specimens, a seemingly medial continuation of the olfactory trigone could be traced towards the septal area and the corpus callosum. According to Allison [18], the medial olfactory stria describes a thin fascicle running from the medial aspect of the olfactory tract to the olfactory tubercle, but it does not reach the septal region of the subcallosal gyrus. Further, Allison [18] pointed out that no significant amount of fibers run from the olfactory tract to the septal area in human. The term medial olfactory stria has been further questioned by Sakamoto et al., stating that the structure described as the medial stria is probably a misinterpretation of a gyrus formed by the olfactory sulcus [36]. Thus, we conclude that the medial striae found in some of our specimens were, in fact, gyri, explaining why they could not be identified in the majority of cases. Our quantitative results show no significant difference between the length and width of the left and right olfactory tracts or between the length of the medial striae in those cases it could be reliably identified. We found a difference in the lengths of the lateral striae, with the left side stria being significantly longer. We acknowledge that the measurements of the striae are technically challenging due to high variability between specimens, and our sample size of five hemispheres per side constitutes a relatively small study population.

We also aimed to elucidate whether the frontal lobe receives direct projections from olfactory bulb via the olfactory tract. Our results support the idea of a novel direct projection from the olfactory tract, thus possibly defining the orbitofrontal cortex as a primary olfactory area. This is in contrast to previous descriptions which define the orbitofrontal cortex as a secondary olfactory area as it receives projections from all primary olfactory areas except from the olfactory tubercle [35,37]. However, the current definition of the primary olfactory areas has been supported by relatively recent imaging studies [14]. We were not able to ensure the same results in our tractographic reconstruction. Direct fibers from the bulb region were constructed as a connection with the frontobasal areas, and the reconstructed fibers were considered as part of UF systems. Thus, we are open to the possibility that our findings during ex vivo dissection were parts of adjacent U-fibers or that the fibers in question represented reciprocal fibers to the orbitofrontal cortex from the primary olfactory cortices through the uncinate fasciculus. Another possibility is that by using the automatic recognition function in DSI studio, some new or unknown connections might be missed or be considered as part of the most known pathways. We conclude that while a strong body of research supports the classic view of the olfactory cortices, our findings of a novel direct projection from the stria to the frontal lobe implies that more research is needed in order to fully understand the anatomical organization of the olfactory system and its projections.

### 4.2. Olfactory System Connectivity: Secondary Olfactory Network

Based on the findings from the dissection procedures and the different levels of connectivity analyzed among the bulb/tract and the primary and secondary olfactory areas, we identified four major pathways possibly supporting a secondary olfactory network. We found the uncinate fasciculus to be represented at all levels of olfactory system connectivity. The uncinate fasciculus is involved in a wide range of functions related to social behavior, such as language processing, episodic memory and in the integration of social and emotional information [38]. Its role has been implied in cognitive improvement in the elderly after olfactory stimulation, and interestingly, temporal lobe epilepsy involving the uncinate fasciculus can present as olfactory hallucinations, a phenomenon termed uncinate seizures [39]. Our findings suggest that the anterior commissure is in a close anatomical relationship with bilateral projections directly from the olfactory bulb, a structural network that has not been previously described in detail. The anterior commissure was present in all levels of olfactory system connectivity, and it is recognized for enabling interhemispheric connections relaying motor and sensory system information. Previous research has described it as an important olfactory pathway between the anterior olfactory nucleus and several parts of the primary and secondary olfactory areas [37]. We also report the involvement of the cingulum connecting the primary and secondary olfactory areas, a finding supported by previous results regarding its involvement in olfactory processing [40]. The cingulum is involved in the limbic system taking part in cognitive function, attention and emotional regulation. Decreased connectivity in the cingulum has been linked to impaired olfactory function in patients with early-stage Parkinsons disease [41]. Finally, the inferior fronto-occipital fasciculus seems to be involved in all levels of olfactory network connectivity, as suggested by previous research [42]. As with the other tracts we report to be involved in olfactory processing, IFOF is involved in complex cognitive processes such as language processing, attention, planning, emotional behavior and reading, as reviewed by Benedicts et al. [43].

No significant lateralization was discerned when analyzing the qualitative parameters of the tracts. Previous research has focused on functional lateralization, but no previous research have looked specifically at lateralization of the connectivity [8,44]. Thus, we suggest further research is needed in order to elucidate any lateralization of connectivity in the olfactory system.

### 4.3. Sensory and Cognitive Implications

The correlation analysis showed no significant correlations between the DTI metrics and odor identification/taste intensity tests. This is expected as our study population is relatively young and healthy, thus representing a homogenous population with normal sensory acuity.

The correlation between cognitive tests and DTI metrics of the secondary cognitive networks showed a positive result (*p* < 0.05, FDR < 0.05) between the axial diffusivity and the mean and radial diffusivity of the right IFOF, and the results of the oral reading recognition test. This is in line with previous research linking IFOF to complex cognitive procedures like reading and language processing [45]. While our current results could not display a correlation between the odor identification test and DTI parameters of the tracts included in our study, we suggest that these results should be viewed as an indicator of the olfactory system overlapping with functional networks associated with cognitive processing and behavior [9,12].

### 4.4. Clinical Implications and Future Perspectives

We report that the olfactory system seems to be organized through different levels of connectivity. The reported direct connection between the olfactory bulb and the secondary olfactory cortices is an important finding pointing towards a need for further re-evaluation of classic olfactory system anatomy. While we do not see significant correlations between olfactory and gustatory sensory acuity and the white matter tracts included in the study, we suggest that further research is warranted in a more varied population to elucidate a possible connection between sensory impairment and DTI parameters of the tracts. The olfactory system has been linked to cognitive improvement through olfactory stimulation, as shown previously [46]. Thus, there is a clear implication for successive studies on olfactory system plasticity in the context of neurorehabilitation [13]. We suggest studies focusing on specific olfactory fiber tracts and the incorporation of fMRI data in order to deepen our understanding of olfactory function in regard to morphological and diffusion-related parameters. We also suggest more studies on possible correlations between olfaction and cognition, with a larger study population and integrated fMRI data. This could provide valuable insights in the role of the olfactory system in cognitive restoration and inspire future research.

### 4.5. Limitations and Methodological Considerations

Our study is the first white matter dissection analysis of the olfactory system. The white matter dissection technique allows for precise and delicate analysis of relatively small tract components, making it suitable for analysis of the olfactory bulb anatomy. It is important to note that the quality and integrity of our specimens varied due to postmortem factors beyond our control. Thus, we cannot exclude that this may have impacted our results. The dissection technique was optimized for micro dissection, with the person performing the procedures being trained by highly skilled professionals in the field. We acknowledge that the study population is relatively small.

In regard to our analysis of the HCP data, it is important to point out that the study was conducted on a healthy cohort. This is crucial in order to establish baseline values for further reference and for a high level of general applicability, but it carries the risk of less pronounced differences in the study population, making it harder to draw clear conclusions. We based our analysis on an atlas-based approach and on a standardized template for spatial normalization. This is crucial for standardization and repeatability, but somewhat limits the possibility for individual variation. We also acknowledge that our study population of 75 individuals is relatively small. It is possible that a larger study population would have yielded less homogenous results in our correlation analyses.

## 5. Conclusions

Thus, we present the first combined fiber dissection analysis and tractography of the olfactory system. We report that the primary olfactory network is constructed by the olfactory tract/bulb connecting primary olfactory cortices through the lateral stria, which seems to be more pronounced on the left side. Major associative white matter pathways such as UF, IFOF and cingulum support the connectivity between primary and secondary olfactory areas together with the anterior commissure. We suggest considering these structures as a secondary olfactory network and that they may therefore represent target structures for olfactory-directed rehabilitation. Further work is needed in order to advance our knowledge of the intricate connections of the system in order to attain deeper understanding of pathological and physiological implications.

## Figures and Tables

**Figure 1 brainsci-15-00368-f001:**
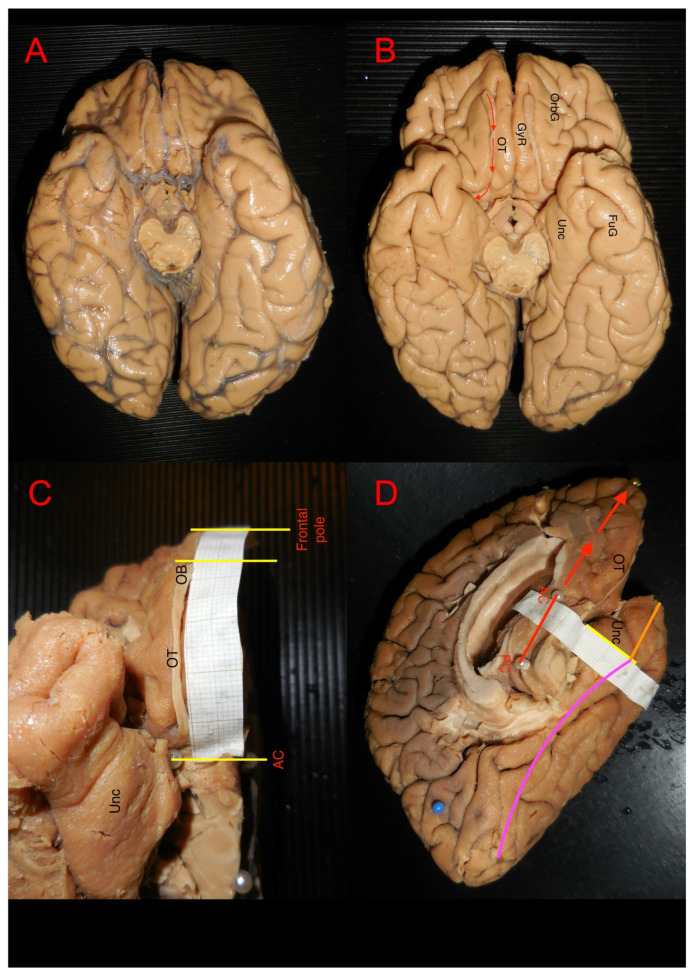
The image shows the first steps of white matter dissection preparation and measurement of cortical/superficial structures. In (**A**,**B**), the ventral point of view of a brain before and after removal of arachnoid layers. In (**C**), the stage of measurement of olfactory tract (OT) and olfactory bulb (OB). In (**D**), the ventro-medial point of view of a left hemisphere during the measurement of cortical size of mesial temporal structures. The red arrows indicate the AC–PC line projection. The orange line indicates the size of the temporal pole from the tip to the perpendicular line crossing the AC point on the midline. The pink line indicates the approximative length and measurement technique for the fusiform gyrus (FuG). (GyR: gyrus rectus; OrbG: orbitofrontal gyrus; Unc: Uncus).

**Figure 2 brainsci-15-00368-f002:**
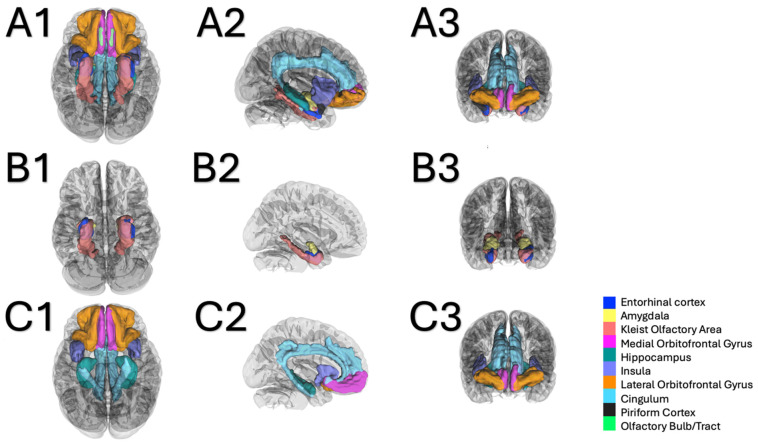
Conceptual representation of the specific regions of interest (ROIs) included in the tractography analyses used in cohort 1. (**A1**) (inferior axial view), (**A2**) (lateral sagittal view) and (**A3**) (frontal coronal view) represent the complete set of ROIs including the olfactory bulb/tract. (**B1**) (inferior axial view), (**B2**) (medial sagittal view, left hemisphere) and (**B3**) (frontal coronal view) represent the ROIs defining the primary olfactory cortices. (**C1**) (inferior axial view), (**C2**) (medial sagittal view, left hemisphere) and (**C3**) (frontal coronal view) represent the ROIs defining the secondary olfactory cortices. Color legend in figure.

**Figure 3 brainsci-15-00368-f003:**
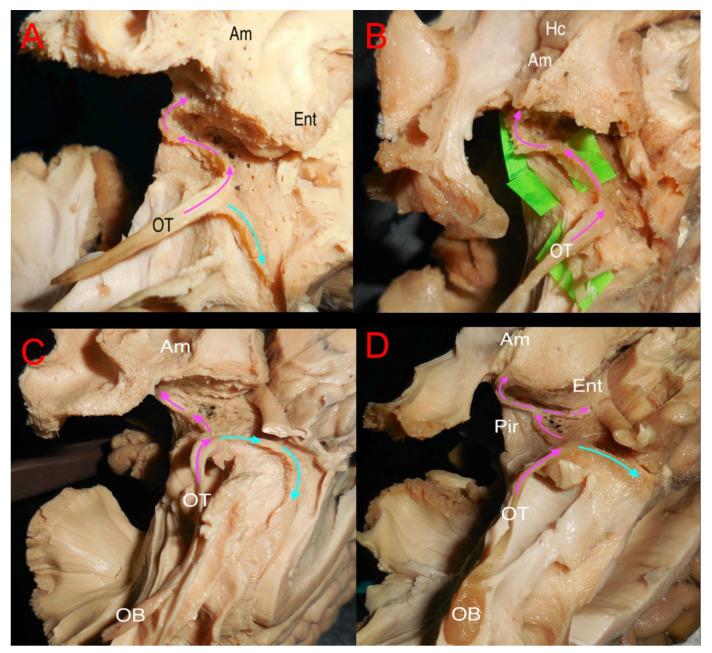
The image shows the dissection of the olfactory tract and the identification of white matter structures forming the lateral and medial strias. In (**A**), ventral anterior point of view of a left hemisphere, showing in pink the direction of the lateral stria to the temporal structures and in light blue the direction of the medial stria from the trigonum. In (**B**), the fibers forming the lateral stria were isolated with the green tags showing an intact pathway. In (**C**,**D**), the light blue arrows show the dissection of the medial stria without a clear white matter structure but a continuity with the cortex of the anterior cingulate/subgenual cortex was displayed. Am: Amygdala; Ent: Entorhinal cortex; OT: olfactory tract; Hc: hippocampus; Pir: piriform cortex; OB: olfactory bulb.

**Figure 4 brainsci-15-00368-f004:**
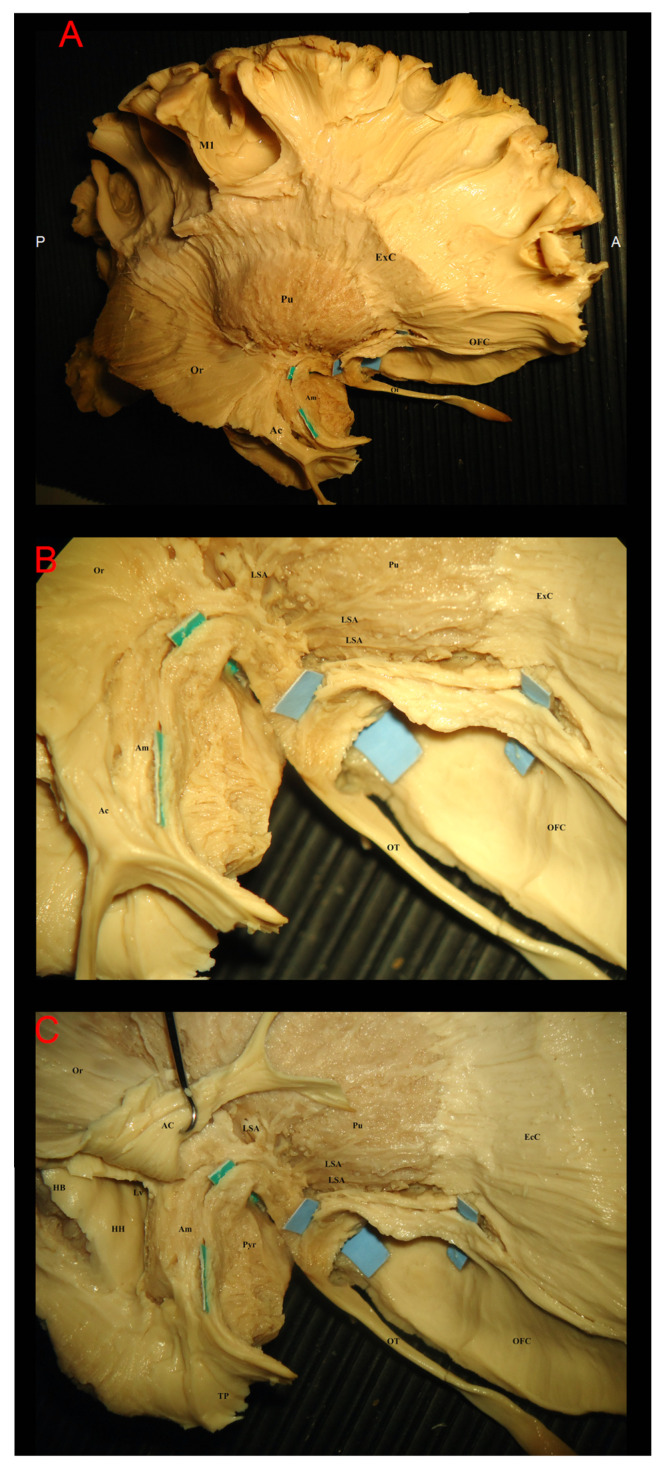
The image shows in the fronto-temporal dissection of the right hemisphere at the level of basal ganglia. In (**A**), a wider point of view showing the relationship between olfactory tract (OT) the putamen and the white matter structures at the level of frontal lobe with orbito-frontal cortex (OFC), external capsule (ExC). On the temporal portion, optic radiation (OR), anterior commissure (AC) and amygdala (Am) are exposed. In (**B**), a detail of the olfactory trigonum from the anterior lateral angle is provided. The green tags show the direct connection between fibers from the olfactory tract and amygdala. A close relationship with the overlapping AC fibers is also visible. The light blue tags in this picture show the continuity of some fibers of the OT to the OFC. Lenticulo-striatal arteries (LSA) are exposed in close proximity to the putamen (Pu) to define the indirect position of the anterior perforating substance. In (**C**), the fibers of the anterior commissure were retracted to show the relationship with the hippocampus head (HH) and hippocampus body (HB) with a medial portion of the temporal pole (TP) left anteriorly and the pyriform cortex (Pyr) medially. M1: primary motor area; A: anterior; P: posterior.

**Figure 5 brainsci-15-00368-f005:**
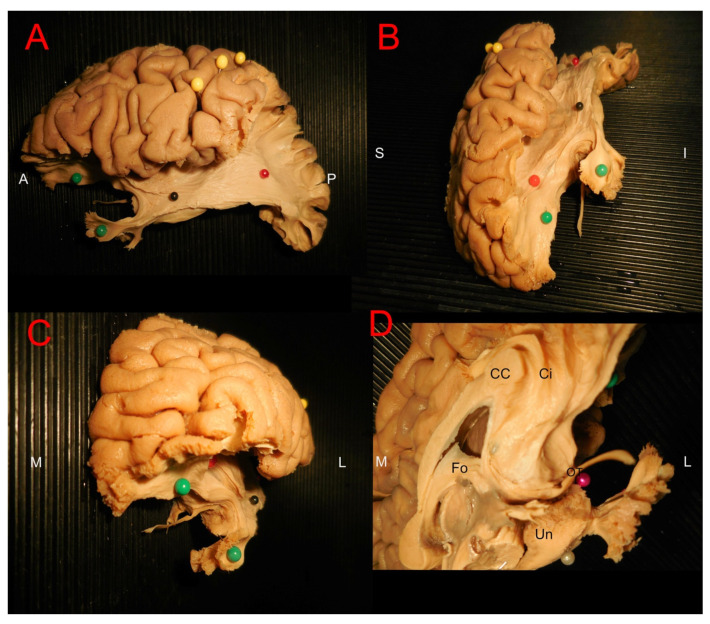
The image shows a fronto-temporal dissection of a left hemisphere with details of associative pathways in relation to the olfactory tract and olfactory regions. In (**A**), the late stage of dissection at the level of the external capsule with yellow pins indicating the central sulcus, the red pin indicating the IFOF, the green pins indicating the course of the uncinate fasciculus and the black pin showing the expected position of the Meyers loop. In (**B**), anterior lateral projection of a later dissection stage showing the olfactory tract and the relationship with UF and the frontal portion of the IFOF (red pin). In (**C**), an anterior (coronal) projection of the same relationship with fibers of the lateral stria reaching the amygdala and the fibers from the UF running dorsally, laterally and ventrally. In (**D**), a medial ventral point of view showing the medial stria reaching the cortex below the anterior cingulate area/subgenual region. A relationship between the Fornix (Fo), cingulum (Ci), corpus callosum (CC) and uncus (Un) is showed. A: anterior; P: posterior; S: superior; I: inferior; M: medial; L: lateral.

**Figure 6 brainsci-15-00368-f006:**
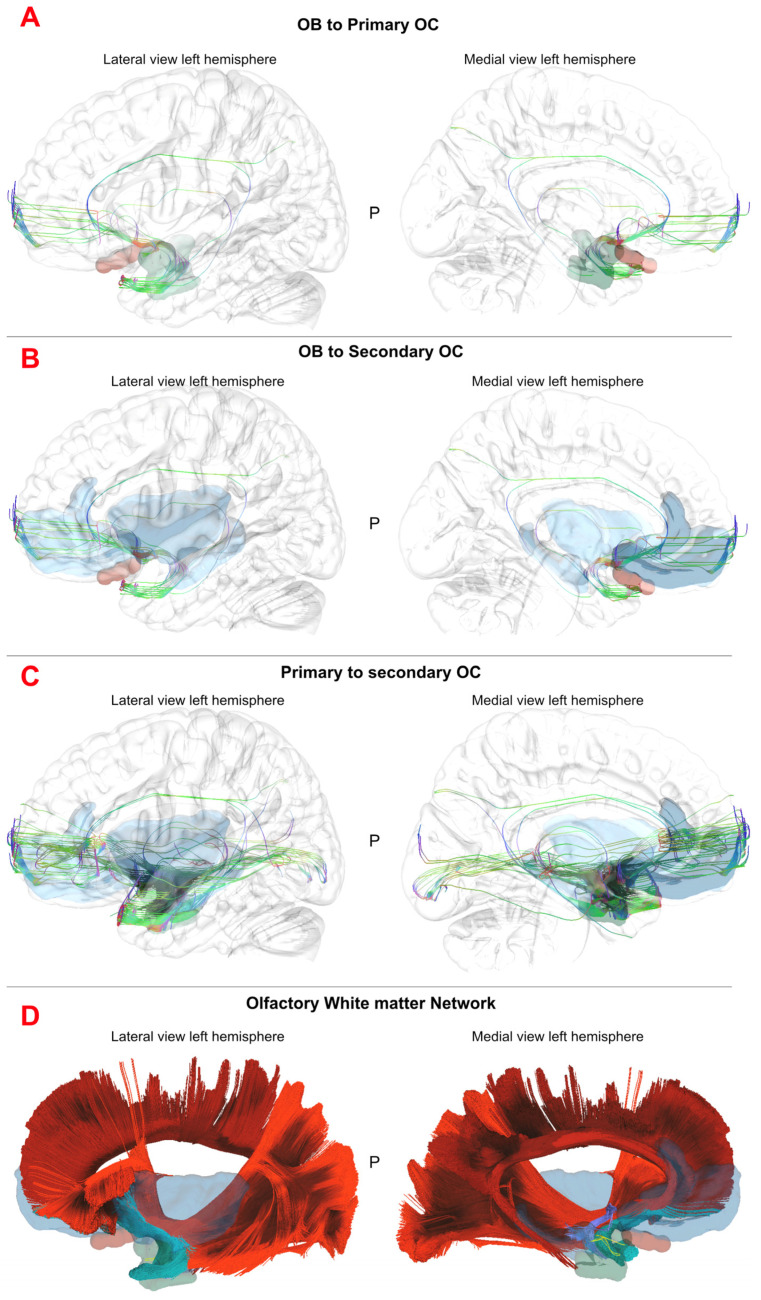
The image summarizes the findings of the DTI analysis on the left-sided hemisphere in MNI space. In (**A**), the lateral and medial projection of the left hemisphere (glass brain) with the olfactory bulb-tract (OB) in red, the primary olfactory cortices (OC) in green and the reconstructed white matter fibers between them. In (**B**), the reconstruction of the connectivity between OB and secondary OC, in light blue. In (**C**), the connectivity between primary and secondary OC is displayed. In (**D**), the image shows the atlas-based reconstruction of the cingulum (dark red), IFOF (light red), uncinate fasciculus (green) and anterior commissure (purple). The lateral stria (yellow) runs anterior and with a medio-lateral direction with respect to the AC fibers and UF fibers. In its temporal portion, it also runs anterior in respect to the temporal horn and therefore the hippocampus. The olfactory bulb, primary and secondary OC are displayed on lateral and medial parts of a left hemisphere to show the anatomical/topographical relationship with the reconstructed white matter bundles. P: Posterior.

**Table 1 brainsci-15-00368-t001:** Mean and range values (in centimeters) of length, width and height of the measured areas. Mean values are presented as means ± SD (given in parentheses). N = 10 in all measurements, except for the width of the frontal lobe where *n* = 8 because the frontal lobe was damaged in two specimens.

Measured Area		Mean Values			Range Values		
		Length	Width	Height	Length	Width	Height
Uncus + parahippocampal gyrus		4.6 (0.4)	2.2 (0.4)		1.6	1.1	
Fusiform gyrus		7.5 (0.7)	1.7 (0.5)		2.1	1.6	
Frontal lobe		7.5 (0.3)	3.4 (0.3)		1.0	1.0	
Olfactory tract	Both	4.2 (0.7)	0.3 (0.1)		1.8	0.2	
	Left	4.2 (0.7)	0.3 (0.1)		1.6	0.2	
	Right	4.3 (0.7)	0.3 (0.1)		1.6	0.1	
Temporal lobe		2.9 (0.3)			1.1		
AC-PC line		2.5 (0.1)			0.4		
Gyrus rectus			0.6 (0.1)			0.4	
Medial orbital gyrus			1.4 (0.3)			0.7	
Medial gyrus rectus + rostral gyrus				2.1 (0.2)			0.7
Lateral stria	Both	2.8 (0.7)			2.4		
	Left	3.3 (0.3)			0.8		
	Right	2.3 (0.6)			1.6		
Medial stria	Both	1.5 (0.7)			2.2		
	Left	1.4 (0.9)			2.2		
	Right	1.5 (0.6)			1.5		

**Table 2 brainsci-15-00368-t002:** A comparison of the left and right hemisphere regarding the length and width of the olfactory tract, as well as the length of the medial and lateral stria by using an unpaired two-tailed *t*-test. *p* < 0.05 is considered significant.

Comparison of Left and Right Side	*p*-Value	Left Mean Value (cm)	Right Mean Value (cm)
Olfactory tract length	1	4.2	0.3
Olfactory tract width	0.2	4.3	0.3
Lateral stria length	0.01	3.3	2.3
Medial stria length	0.8	1.4	1.5

**Table 3 brainsci-15-00368-t003:** Explorative tractographic analysis of three levels of olfactory system connectivity presented as mean % of reconstructed pathways according to the automatic recognition function from DSI Studio. The main tracts representing first-level (bulb/tract to primary olfactory cortices) connectivity are the UF, the AC and the IFOF. The main tracts representing second-level connectivity (bulb/tract to secondary olfactory cortices) are the UF, the AC, the IFOF and the cingulum. The main tracts representing tertiary-level (primary olfactory cortices to secondary olfactory cortices) are the UF, the IFOF, the AC and the cingulum.

Tract	Olfactory Bulb/Tract to Primary OC, Mean %	Olfactory Bulb/Tract to Secondary OC, Mean %	Primary to Secondary OC, Mean %	Mean FA	Mean RD
Right UF	71.4	53.8	42.6	0.18898	0.13288
Left UF	70.5	66.9	44.1	0.16306	0.1588
AC	56.4	57.3	23.2	0.3420	0.2238
Right IFOF	31.6	30.6	25.3	0.3709	0.2160
Left IFOF	28.9	21.5	25.1	0.2648	0.2016
Right Cingulum		66.5	29.8	0.3294	0.2762
Left Cingulum		65.4	27.6	0.5620	0.1739

## Data Availability

Post-processed MRI data from the HCP database that support the findings are available on request from the corresponding author. The donor-related data are not publicly available due to privacy or ethical restrictions.

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
