# Peer review of "The Bulb, the Brain and the Being: New Insights into Olfactory System Anatomy, Organization and Connectivity"

_brainsci, 2025, doi:10.3390/brainsci15040368_

Round 1

Reviewer 1 Report

Comments and Suggestions for Authors

The manuscript discusses an important research question. There are some comments to improve the manuscript

Please define all abbreviations  before use. Try to minimize the abbreviations in the abstract.

The abstract is wordy and lacks numerical results.

The introduction is clear and the objectives are well stated.

Please use more subheadings in the methods. The design and ethical approval were under (white matter dissection) section, Please use a separate  subheading.

Details of statistical analysis (such as t-test) should be added in the analysis section.

In table 2, include the values of the right and left side.

References list need an update with recent references.

Author Response

Comments 1: The manuscript discusses an important research question. There are some comments to improve the manuscript

Comments 2: Please define all abbreviations  before use. Try to minimize the abbreviations in the abstract.

Response : The abstract has been clarified and the abbreviations have been written out for clarity. Line 14, 15, 25, 30, 31. 

Comments 3: The abstract is wordy and lacks numerical results.

Response: Thank you for pointing this out. The scope of the article requires a somewhat wordy abstract. Some minor adjustments have been made to clarify the abstract.

Comments 5: Please use more subheadings in the methods. The design and ethical approval were under (white matter dissection) section, Please use a separate  subheading.

Response: Thank you. Some of the subheadings have been revised and a new subheading has been added for clarity. See lines 83-90. New subheadings on lines 114 and 122.

Comment 6: Details of statistical analysis (such as t-test) should be added in the analysis section

Response 6: The details of the statistical analysis of white matter dissection data were revised and relocated.

Comments 7: In table 2, include the values of the right and left side.

Response: The values of the right and left sides are added to Table 2. 

Comments 8: References list need an update with recent references.

Response: Thank you for the suggestion. We have now added several references describing the organization of olfactory systems, the implication of function loss in the connectivity and anatomical analysis, and the effect of olfactory functions on well-being and general health.

Reviewer 2 Report

Comments and Suggestions for Authors

Overview of the manuscript
The manuscript focuses on investigating the organization and connectivity of the olfactory tract in human. The authors use dissection technique, white matter reconstruction algorithm and 
 DTI metrics to identify major associative, projection and commissural pathways. The authors found that the dissection showed that the lateral stria was significantly longer  on the left side projected towards the amygdala, the entorhinal and piriform cortex. No significant correlation was found between DTI metrics and sensory or cognition test results.

GENERAL COMMENT
The work is interesting in proposing a detailed analysis of olfactory tract in human that nowatoday is influenced by several information coming from animals that are not verified in human.
The work is well performed and the iconographic plan is adequate to show the methodological approach and the results.
Make sure that all acronyms in the text be explained.SPECIFIC COMMENTS
No concerns

Author Response

Comment 1: The manuscript focuses on investigating the organization and connectivity of the olfactory tract in human. The authors use dissection technique, white matter reconstruction algorithm and 
 DTI metrics to identify major associative, projection and commissural pathways. The authors found that the dissection showed that the lateral stria was significantly longer  on the left side projected towards the amygdala, the entorhinal and piriform cortex. No significant correlation was found between DTI metrics and sensory or cognition test results.

Comment 2: The work is interesting in proposing a detailed analysis of olfactory tract in human that nowatoday is influenced by several information coming from animals that are not verified in human.
The work is well performed and the iconographic plan is adequate to show the methodological approach and the results.
Make sure that all acronyms in the text be explained.

Response: Thank you. We have explained the acronyms used in the text. 

Reviewer 3 Report

Comments and Suggestions for Authors

First of all, I would like to express my gratitude for the opportunity to review this manuscript.

The topic is very relevant, since the sense of smell is a powerful stimulus of emotions and memory for humans in particular and animals in general. In addition, aromatherapy has recently become widespread in the treatment and rehabilitation of patients with pathologies of the autonomic nervous system with neurotic disorders and cognitive impairments. Undoubtedly, further advanced research in the field of studying anatomical structures and their pathways opens up more opportunities for finding new ways to modulate the olfactory analyzer in the therapeutic field.

The introduction is very interestingly written, providing detailed information on the anatomy and physiology of the olfactory nerve and the olfactory subcortical and cortical centers. However, the introduction seems short to me, and could you add some recent information about the use of nerve stimulation in medicine or, conversely, about the effect of irritating odors on the deterioration of human health?

Add reference, please, to line 43, 68,

Can you describe in detail the five factor model of personality mentioned in line 53?

Please describe in detail the abbreviations HCP and DTI mentioned in lines 76-78!

The purpose of the study requires further reformulation with greater specificity.

In Materials and Methods: Please indicate the country where the study was conducted. The “Department of Medical Cell Biology, Uppsala University”.

Line 83 states that the sample was taken from 8 cadavers. However, the study was only conducted on five right-sided and five left-sided hemispheres of the human brain. Please explain!

Overall, the materials and methods are described in detail, including white matter dissection, HCP data, fiber tracing analysis, in vivo exploratory white matter analysis, regional-based analysis and atlas-based analysis, in plain language, step by step. Moreover, the preparation of anatomical materials was carried out in accordance with the latest scientific achievements in this field.

In my opinion, Figure 1 is very relevant and informative and clearly describes the first steps of preparation of white matter dissection and measurement of cortical/superficial structures of the olfactory tract and bulb. Please add "olfactory tract and bulb" to the title of the figure.

Figure 2 is very informative and makes it easier for the reader to understand this section. Please provide the full words of the abbreviation “ROI” in title.

The results of the study are carried out in accordance with the set tasks with detailed explanations. On anatomical preparations, recorded in figures 3-5. The obtained data do not raise doubts about their reliability. However, the small sample size may miss rarer anatomical variations.

In Figure 6, how can you differentiate the hippocampal white matter fibers from the olfactory white matter fibers?

The discussion is described in accordance with the obtained results with references to the latest discoveries in this area. Undoubtedly, the authors highlight the most important results and emphasized the novelty of their research.

In conclusion authors  present the first combined fiber dissection analysis and tractography of the olfactory system. Based on obtained results authors concluded that the primary olfactory network is constructed by the olfactory tract/bulb connecting primary olfactory cortices through the lateral stria, which seems to be more pronounced on the left side. Major associative white matter pathways such as UF, IFOF and cingulum are supporting the connectivity between primary and secondary olfactory areas together with the Anterior commissure.  In results authors suggest considering these structures as a secondary olfactory network and that they may therefore represent target structures for olfactory-directed rehabilitation. Further work is needed in order to advance our knowledge of the intricate connections of the system in order to attain deeper understanding of pathological and physiological implications.

I really enjoyed the entire article. Thanks to the authors for their excellent work in presenting this research. However, there are some comments that need to be answered.

Author Response

Comment 1: The introduction is very interestingly written, providing detailed information on the anatomy and physiology of the olfactory nerve and the olfactory subcortical and cortical centers. However, the introduction seems short to me, and could you add some recent information about the use of nerve stimulation in medicine or, conversely, about the effect of irritating odors on the deterioration of human health?

Response: We appreciate the suggestion. We have now added more references and a short paragraph on olfactory nerve stimulation and rehabilitation and the effect of irritating odors on the deterioration of human health and cognitive functions. We aimed anyway to not have a very long introduction to go straight to the aim, considering the nature of the special issue and similar articles on the topic.

Comment 2: Add reference, please, to line 43, 68,

Response: Reference has been added to line 43 and the wording on line 68 was changed as to not be speculative.

Comment 3: Can you describe in detail the five-factor model of personality mentioned in line 53?

Response: An explanation has been added to the paragraph. 

Comment 4: Please describe in detail the abbreviations HCP and DTI mentioned in lines 76-78!

Response 4: The abbreviations has been written out. 

Comment 5: The purpose of the study requires further reformulation with greater specificity.

Response 5: The purpose has been reformulated in the abstract.

Comment 6: In Materials and Methods: Please indicate the country where the study was conducted. The “Department of Medical Cell Biology, Uppsala University”.

Response 6: City and country has been added to the text.

Comment 7: Line 83 states that the sample was taken from 8 cadavers. However, the study was only conducted on five right-sided and five left-sided hemispheres of the human brain. Please explain!

Response. We understand the difficult interpretation of the results. Only intact hemispheres were used for this study. If during the extraction of the hemispheres cortical or olfactory nerve damages were detected the damaged hemisphere was not included in this study. In summary, it is a technical reason, but all the donors were healthy and not deceased from neurological disease.

Comment 8: In my opinion, Figure 1 is very relevant and informative and clearly describes the first steps of preparation of white matter dissection and measurement of cortical/superficial structures of the olfactory tract and bulb. Please add "olfactory tract and bulb" to the title of the figure.

Response 8: Olfactory tract and bulb has been added to the figure text of figure 2.

Comment 9: Figure 2 is very informative and makes it easier for the reader to understand this section. Please provide the full words of the abbreviation “ROI” in title.

Response 9: The abbreviation has been described in full in the title of Figure 2.

Comment 10: The results of the study are carried out in accordance with the set tasks with detailed explanations. On anatomical preparations, recorded in figures 3-5. The obtained data do not raise doubts about their reliability. However, the small sample size may miss rarer anatomical variations.

Response. Thanks for the comment. We agree with the limitation of the small sample size and with the need of further studies with larger sample numbers. We also added this comment to the limitation section.

Comment 11: In Figure 6, how can you differentiate the hippocampal white matter fibers from the olfactory white matter fibers?

Response: In the image, the olfactory fibers start from the frontobasal area close to the bulb and continue to the temporomesial lobe and amygdala region but are more anterior than hippocampal fibers ( on sagittal view).

The discussion is described in accordance with the obtained results with references to the latest discoveries in this area. Undoubtedly, the authors highlight the most important results and emphasized the novelty of their research.

In conclusion authors  present the first combined fiber dissection analysis and tractography of the olfactory system. Based on obtained results authors concluded that the primary olfactory network is constructed by the olfactory tract/bulb connecting primary olfactory cortices through the lateral stria, which seems to be more pronounced on the left side. Major associative white matter pathways such as UF, IFOF and cingulum are supporting the connectivity between primary and secondary olfactory areas together with the Anterior commissure.  In results authors suggest considering these structures as a secondary olfactory network and that they may therefore represent target structures for olfactory-directed rehabilitation. Further work is needed in order to advance our knowledge of the intricate connections of the system in order to attain deeper understanding of pathological and physiological implications.

Round 2

Reviewer 3 Report

Comments and Suggestions for Authors

Thanks again for the opportunity to review this manuscript again.

The topic is very relevant due to the lack of advanced research in studying anatomical structures of olfactory nerve and its pathways. Authors first  combined fiber dissection analysis and tractography of the olfactory system and proven that the primary olfactory network is defined by the olfactory tract/bulb and primary olfactory cortices through the lateral stria only. Moreover, authors reported that the uncinate fasciculus, inferior fronto-occipital fasciculus and cingulum are the associative pathways supporting the connectivity between primary and secondary olfactory areas together with the anterior commissure

In my opinion, the changes and additional information in the introduction made this section more advanced and scientific.

The change in the purpose made it more specific and understandable, as well as relevant to the research conducted and the results obtained.

Overall, the materials and methods are described in detail

The results are presented clearly using explanatory tables and illustrations, making the reader's task easier.

The corrections in figures 1, 2 made them more informative and understandable for readers

The discussion and conclusion are focused on the objectives set and correspond to the results obtained.

The authors convincingly answered all my questions and left me with no doubts about the scientific nature, novelty and reliability of this work.